# Cancer Cells Resistance Shaping by Tumor Infiltrating Myeloid Cells

**DOI:** 10.3390/cancers13020165

**Published:** 2021-01-06

**Authors:** Marcin Domagala, Chloé Laplagne, Edouard Leveque, Camille Laurent, Jean-Jacques Fournié, Eric Espinosa, Mary Poupot

**Affiliations:** 1Centre de Recherches en Cancérologie de Toulouse, Inserm UMR1037, 31037 Toulouse, France; marcin.domagala@inserm.fr (M.D.); chloe.laplagne@inserm.fr (C.L.); edouard.leveque@inserm.fr (E.L.); Laurent.Camille@iuct-oncopole.fr (C.L.); jean-jacques.fournie@inserm.fr (J.-J.F.); eric.espinosa@inserm.fr (E.E.); 2Université Toulouse III Paul-Sabatier, 31400 Toulouse, France; 3ERL 5294 CNRS, 31037 Toulouse, France; 4IUCT-O, 31000 Toulouse, France

**Keywords:** microenvironment, resistance, myeloid cells, cancer development

## Abstract

**Simple Summary:**

The tumor is a complex system that is composed of tumor cells, themselves surrounded by many other different cell types. Among these cells, myeloid cells have to eliminate cancer cells to reduce tumor size, but they are also able, depending on the tumor stage, to favor tumor development. Therefore, different cellular interactions and soluble factors that are produced by all these cells can participate to maintain tumor cell survival and favor their proliferation, migration, and resistance to cytotoxic immune cells and therapies. This revue aims to detail the physiological function of myeloid cells, their pathological function, and how they shape tumor cells to be resistant to apoptotic, to immune effector cells, and to therapies.

**Abstract:**

Interactions between malignant cells and neighboring stromal and immune cells profoundly shape cancer progression. New forms of therapies targeting these cells have revolutionized the treatment of cancer. However, in order to specifically address each population, it was essential to identify and understand their individual roles in interaction between malignant cells, and the formation of the tumor microenvironment (TME). In this review, we focus on the myeloid cell compartment, a prominent, and heterogeneous group populating TME, which can initially exert an anti-tumoral effect, but with time actively participate in disease progression. Macrophages, dendritic cells, neutrophils, myeloid-derived suppressor cells, mast cells, eosinophils, and basophils act alone or in concert to shape tumor cells resistance through cellular interaction and/or release of soluble factors favoring survival, proliferation, and migration of tumor cells, but also immune-escape and therapy resistance.

## 1. Introduction

Nowadays, tumor microenvironment (TME) is recognized as an essential element of tumor development and progression. It not only remains in constant contact with the tumor, but it also mediates complex dialog between malignant cells and surrounding tissues. The cellular components of this dynamic network are represented by normal and tumoral tissue-resident cells with a large proportion of recruited immune cells alongside: fibroblasts, neuroendocrine, adipose, endothelial, and mesenchymal cells [1]. All of the cellular and molecular actors of the TME are involved in carcinogenesis through the promotion of tumor: growth, dormancy, invasion, and metastasis. The infiltrating immune cells can be represented by lymphoid cells, such as: CD8, CD4, and γδ T lymphocytes, B cells, and natural killer (NK) cells, and myeloid cells, such as: monocytes/macrophages, dendritic cells (DC), neutrophils, myeloid-derived suppressor cells (MDSC), basophils/eosinophils, and mast cells. In the initial states of oncogenesis, all of these cell populations can help in the elimination of mutated cells. However, after the tumor dormancy and editing phase, the loss of oncoantigens and MHC lead to the immune escape, allowing for further tumor development [2]. TME, including immune cells, is then modified to actively support and promote cancerogenesis and shape the character of emerging tumors [3]. This review aims at summarizing the role of the tumor infiltrating immune cells and, particularly, myeloid cells shaping cancer cells resistance to apoptosis, immune response, and therapy. Following the text, the readers can refer to the figures that resume the role of the different tumor-associated myeloid cells in cancer cells survival, proliferation, and migration (Figure 1), and in cancer cells immune-escape and therapy resistance (Figure 2).

## 2. Macrophages

Macrophages in cancer represent a major part of the immune cells within the TME and they are more frequently associated with a bad prognosis. Understanding the origin and physiological roles of macrophages provides improved insight into their role in cancer.

### 2.1. Origin and Physiological Roles of Macrophages

It was recently shown that most resident macrophages (MPs) in normal tissues are not only derived from bone marrow (BM) progenitors, as previously thought, but also from yolk sac or foetal liver and they are maintained by self-renewal [4,5]. However, during adult life, the rate of resident MPs can also be maintained by the infiltration of blood-derived MPs, except for microglia in the brain [6]. Therefore, monocytes from blood and bone marrow are able to infiltrate tissues and differentiate into specific tissue MPs, but whether they can be considered to be tissue-resident MPs is still a debate. In the blood, two types of monocytes can be distinguished by the expression of the Ly6C marker in mice, Ly6C^+^ monocytes being originate from bone-marrow, and Ly6C^−^ from circulating Ly6C^+^ monocytes [7]. If Ly6C^+^ monocytes respond to damage by infiltrating and differentiating in MPs in the tissues, Ly6C^−^ monocytes remain in the vessels to detect and remove damaged endothelial cells [8]. Therefore, the ratio of these two monocytes populations in the blood can change, depending on different stimuli, including external stimuli. In human blood, three main monocytes populations have been identified, a classical one, CD14^++^CD16^−^, a non-classical one, CD14^int^CD16^+^, and an intermediate one, CD14^+^CD16^+^ [9]. These two latter populations are differentiated from a uniform population CD14^++^CD16^−^ egressing from the bone marrow [10].

Resident macrophages display key roles in growth, tissue development, homeostasis, and remodelling, and they have site-specific phenotypes and functions [11]. It was proposed that the specialization of the resident MPs takes place inside the target tissue, due to close contacts with tissue-specific cells as well as to soluble factors in the tissue environment [12]. Microglia, in the brain and spinal cord, contribute to synaptic maturation during brain development and the clearance of immature or defective neuronal synapses [13]. In the lung, alveolar MPs mediate approximately 30% of the surfactant lipid metabolism. Langerhans cells with cutaneous MPs in the skin are specialized in the formation of the extracellular matrix and in skin layer differentiation. Cardiac-resident MPs are required during heart development and they take part in the regulation of the cardiac rhythm [14]. Kupffer cells in the liver are involved in the modulation of metabolism in hepatocytes, preventing the pathogenic accumulation of lipids [15]. Tissue-resident MPs that are located in the red-pulp region of the spleen have important functions in iron processing connected with the clearance of damaged red blood cells and the erythropoiesis [16]. Besides, MPs from the white-pulp region phagocytose lymphocytes avoiding B cell accumulation and auto antibody production [17].

Apart from physiological functions, monocytes/MPs also display pathological functions in infection/inflammation contexts, tissue repair, and cancer. 

### 2.2. Pathological Functions of Macrophages

MPs are very plastic cells that are able to respond to molecular or cellular signals from the tissue environment. The molecular signals can be endocrine or paracrine signals that originate from phagocytosed cells or microorganisms and from the extracellular matrix/proteins. MPs can also directly interact with other tissue-resident cells, such as immune cells recruited during injury. Indeed, monocytes and MPs are recruited from the bone marrow to the tissue injury site via the chemoattractant CCL2 that is secreted by resident MPs, endothelial cells, myocytes, and fibroblasts [18]. The CCL2 receptor, CCR2, is highly expressed by Ly6C^+^ mouse monocytes [19]. In humans, classical monocytes (CD14^+^CD16^−^) display a high expression of CCR2 and they are involved in responses to bacterial infection and inflammation, in inflammasome signalling, and in low density lipoprotein uptake. In contrast, non-classical monocytes (CD14^int^CD16^+^) display a high expression of genes that are involved in cytoskeletal dynamics, tissue invasion during inflammation and genes suggesting terminal differentiation and cellular maturity [20].

Therefore, monocytes-derived and tissue-resident MPs colocalize to take part in healing and then to the resolution, thanks to the production of cytotoxic and pro-inflammatory mediators, the clearance of invading microorganisms, or removal of apoptotic and damaged cells [21]. On the arrival at the injury site, blood-derived MPs can adopt a pro-inflammatory/M1/classical or anti-inflammatory/M2/alternative phenotype, depending on the cytokines that are present in the microenvironment [22,23]. Regarding the plasticity of MPs, although the framework of M1/M2 polarization is a very useful system for in vitro studies, it is unclear how similar clear-cut phenotypes can be appended during in vivo injury and repair [24]. This M1/M2 paradigm is well pictured by the high expression of M2 markers by tissue-resident MPs when compared to the mature phenotype of monocyte-derived MPs [25,26]. Early on after damage or injury, infiltrated Ly6C^+^ monocytes scavenge apoptotic debris or pathogens or infected cells thanks to the expression of pattern recognition receptors (PRR), Toll like receptors (TLR), scavenger receptors, and Fc receptors that, respectively, recognize microbial antigens, danger signals, or immunoglobulins. These events lead to the activation of transcription factors such as interferon (IFN) regulatory factors and nuclear factor kappa B (NFκB), inducing an M1 polarization and initiation of the inflammatory response [27]. These blood-derived M1 MPs then release pro-inflammatory cytokines (e.g., IL-1β, IL-6, IL-12, IL-23, and TNF-α) and type-1 cell-attracting chemokines (e.g., CXCL9 and CXCL10), favouring the recruitment of more macrophages and leucocytes to help with injury resolution [28]. Once the acute injury has been resolved, MPs are in charge of suppressing inflammation and initiating wound repair. After clearing debris, MPs produce growth factors and mediators, which abrogate the pro-inflammatory function of T cells and other immune cells [29]. This is accompanied by a progressive repolarization of the blood-derived MPs towards a phenotype and functions that are increasingly similar to those of homeostatic tissue-resident MPs [30]. 

Specific plasma membrane receptors induce pro- and anti-inflammatory pathways in MPs. If IFNγ mediates the classical/M1 activation with upregulation of MHCII antigens, induction of nitric oxide synthase (i-NOS) and the production of pro-inflammatory molecules, interleukin-4 and -13 (IL-4 and IL-13) induce the alternative/M2 phenotype that is characterized by the upregulation of CD206, transglutaminase 2, arginase, and the production of IL-10 and chemokines, such as CCL17, CCL22 and CCL24 [28,31]. Like other immune cells, specific functions of MPs are, therefore, coupled to specific phenotypes, even when considering their plasticity, MPs can display intermediate phenotypes in certain inflammatory diseases and cancer [32,33].

### 2.3. Macrophages in Cancer

MPs in cancer are called tumor associated macrophages (TAM) and they represent the major immune component of the TME. According to oxygen ratio and tumor progression, TAM display either a M1 or M2 phenotype. They play a major role in tumor growth, metastatic dissemination, and therapy failure, promoting angiogenesis and secreting different factors that are involved in extracellular matrix (ECM) remodelling that facilitate tumor cell motility and intravasation. High TAM infiltration is generally correlated with poor outcomes in several types of cancer.

#### 2.3.1. Origin and Functions of TAM

Until recently, TAM were considered to exclusively originate from blood-derived MPs undergoing differentiation upon tissue infiltration in response to chemokine and growth factors that are produced by stromal and tumor cells in the TME. Colony-stimulating factor 1 (CSF-1), vascular endothelial growth factor A (VEGF-A), and different CCL (2, 18, 20) were found to act as chemotactic molecules in various cancer [34,35,36]. However, evidence shows that tissue-resident MPs can coexist in tumors with blood-derived MPs and their phenotype can rapidly evolve, depending on the stage of the tumor and the characteristics of the molecular and cellular actors in the TME. 

In early stage tumor development, IFN-α polarizes resident MPs towards an M1 phenotype and activates the infiltration of blood derived-M1 MPs. These MPs directly phagocytize tumor cells expressing low levels of the “don’t eat me” signal CD47, release pro-inflammatory factors that activate Th1 and Th17 immune responses and can also produce TNF-related apoptosis-inducing ligand (TRAIL) that result in TRAIL-induced cancer cell apoptosis [37,38]. In contrast, in more advanced tumors, TAM are polarized to an M2 related phenotype. This polarization occurs, thanks to anti-inflammatory mediators that are produced by the tumor cell itself and by stromal and immune cells in the TME, but also by MPs themselves. CSF-1, CCL2, 3, 14, and IL-4 are common tumor-derived factors driving the recruitment, proliferation, and M2-polarization of MPs [39,40,41]. Other factors are more specific to the type of cancer, such as prostate cancer-derived cathelicidin-related antimicrobial peptide [42] or hypoxic cancer cell-derived cytokines Oncostatin M and Eotaxin [43]. IL-4 can also be secreted in the TME by Th2-polarized CD4 cells as well as IL-10 or IL-13, which lead to STAT-6 activation [44,45,46]. Besides, migration-stimulating factor (MSF), IL-4, and CXCL12 can be secreted by MPs to promote self-polarization [47,48,49]. Finally, hypoxia-inducible factors (HIF-1α and 2α), high-mobility group box 1 protein (HMGB1), extracellular ATP, or tumor-derived ECM components are also potential factors that promote M2 polarization [50,51,52].

#### 2.3.2. How Can TAM Shape Cancer Cell Resistance?

M2-like MPs in TME are highly involved in cancer cell resistance by promoting cancer initiation, angiogenesis, the establishment of a premalignant niche, metastasis, and immune suppression.

It has been shown that an inflammatory microenvironment promotes genetic instability, leading to the proliferation of epithelial cells, but also the infiltration of immune cells, such as macrophages. On site, TAM can secrete IL-23 and IL-17, which promote cancer cell proliferation. IL-23 signaling in tumor cells is important for the intra-tumoral production of downstream cytokines, which are either direct (IL-6, IL-22) or indirect (IL-17A) STAT3 activators [53]. IL-6 that is produced by TAM also promotes tumor cells proliferation and invasive potential via STAT3 signaling [54]. TAMs also represent a strong source of iron, which is essential in tumor cell division, growth, and survival, and motility through the remodeling of the extracellular matrix [55]. 

TNF-α, which is a key player in NFκB upregulation, is produced in the TME by, amongst others, TAM and it induces migration and invasion potential of cancer cells [56]. Cancer cells motility is also favored by various metalloproteinases (MMPs) and cathepsins that are produced by TAM activated by TGF-β in the TME [57,58]. Some chemokines, such as CCL18 produced by TAM, also promote the migration of cancer cells and metastasis through the clusterization of integrins [59]. A mouse study showed that mouse MPs-derived insulin growth factor 1 (IGF-1) induces the migration of epithelial ovarian cell lines [60]. Finally, it has been shown that MPs-derived microRNA (miR-223) also regulate tumor invasion [61].

TAM also take part in the promotion of the formation of blood vessels within the tumor providing nutrition for tumor growth [62]. Several pro-angiogenic factors, such as TGF-β, VEGF, PDGF, and angiogenic chemokines, are produced by TAM in the TME. CCL18 produced by TAM promote, synergically with VEGF, the endothelial cell migration and angiogenesis [63]. Other chemokines, such as CXCL1, 8, 12, 13, and CCL2, 5 produced by TAM, help with the angiogenesis switch in tumor tissues [28]. Finally, TAM can be found in hypoxic parts of the tumor and it can express HIF-1α, which regulates the transcription of VEGF largely associated with angiogenesis [58]. 

The epithelial-mesenchymal transition (EMT) is a fundamental process for tumor progression and metastasis, during which TAM plays an active role though interactions with tumor cells and the production of facilitating factors of EMT. Through the production of EGF, TAM can induce the EMT of cancer cells by activating the EGFR/ERK1/2 signaling pathway [64].

TAM are also able to protect tumor cells from immune attacks, inhibiting T cell proliferation, function, and recruitment through the release of immunosuppressive cytokines. They are able to neutralize the recruitment and functions of cytotoxic CD8 T-cells and natural killer cells through the secretion of IL-10 and TGF-β in the TME [65,66]. TAM-derived TGF-β also decreases antigen presentation, which reduces DC migration and increases apoptosis [67]. On the contrary, the production of CCL17 and CCL22 by TAM promotes the infiltration of Th2 and Treg populations in tumors [68]. TAM-derived prostaglandin E2 (PGE2), IL-10, and indoleamine 2,3-dioxygenase (IDO) play important roles in the suppression of cytolytic T lymphocytes and in the induction of Treg function [69,70]. IL-10, alone or in concert with IL-6, causes the upregulation of macrophage B7-H4 expression, which is responsible for the suppression of tumor-associated antigen-specific T cell immunity [71]. Moreover, IL-10 and PGE2 can induce the expression of immune-checkpoint ligands (PD-L1) in myeloid cells, which can inhibit cytolytic T lymphocytes responses [72,73]. Finally, IL-10 acts in an autocrine circuit in TAM in order to restrain their expression of IL-12 and also inhibits the release of IFN-γ [48].

The resistance of tumor cells to cytotoxic T cells can also be induced by reactive oxygen species (ROS) and reactive nitrogen species (RNS) that are produced by TAM through iNOS and arginase I, two enzymes that are very active in TAM [74].

TAM can also be responsible for tumor resistance to treatments. TAM-derived exosomes have been shown to be involved in mediating the resistance of gastric cancer cells to cisplatin [75]. Endocrine resistance in breast cancer cells can be increased by TAM-derived CCL2 through the activation of the PI3K/Akt/mTOR signaling pathway [40]. Autophagy in hepatocellular carcinomas cells can be induced by TAM, leading to oxaliplatin resistance [76]. It was recently shown that depletion of TAM by an anti-CSF-1R enhanced the anti-tumor effect of docetaxel in a murine epithelial ovarian cancer [74]. In Chronic Lymphocytic Leukemia (CLL), the nurse like cells (NLC), which are the specific protumoral TAM of the CLL reside in lymph nodes, spleen and bone marrow where they protect CLL B cells against apoptosis but also against chemotherapies such as ibrutinib [77,78]. This protection has been shown to depend on cell contact and soluble factors that are produced by TAM and, in an autocrine manner, by CLL B cells themselves [79,80,81,82].

Anticancer immunotherapies may also be reduced by TAM when their suppression of TME correlates with an increase of DC-vaccination therapy in a malignant mesothelioma mouse model [83] or an increase of anti-PD1 treatment favoring CD8 T cells recruitment to the tumor site [84,85]. Finally, high levels of TAM in the TME were also shown to increase the resistance of tumor cells to Vascular-targeted photodynamic: paldeliporfin (VTP) therapy in a prostate mouse model [86].

## 3. Dendritic Cells

Dendritic cells (DC) are derived from bone marrow progenitors and they can infiltrate the TME. They represent a small fraction in the TME, but they can play a key role in the sensing of infiltrated T cells or have an immunosuppressive role, leading to the tumor resistance.

### 3.1. Origin and Roles of Dendritic Cells

DC represent the regulators of innate and adaptive immune responses. These cells are able to present antigens to T lymphocytes. Several subsets of DC can be distinguished, depending on the tissue, e.g., classical/conventional DC (cDC) and plasmacytoid DC (pDC). Amongst cDC, the cDC1 present exogenous cell-associated antigens to CD8+ T cells, regulating their cytotoxic responses to intracellular pathogens [87] and cancer cells. In addition, cDC1 regulate innate immune responses through the production of IL-12 to activate the expression of IFN-γ by NK cells [88]. The cDC2, which present soluble antigens to CD4 T cells, are involved in the regulation of responses to extracellular pathogens, parasites, and allergens [89]. As for pDC, they rapidly respond to pathogens thanks to their expression of TLR7 and TLR9, which recognize single-stranded RNA and CpG dinucleotides, respectively. Through the production of type I and II interferons, these cells regulate the expansion of NK cells and CD8 T cells, but they also induce the maturation of cDC1 [90]. cDC and pDC derive from the common DC progenitor by a differentiation that is strictly restricted to this hematopoietic lineage. The common DC progenitor derive from the differentiation of hematopoietic stem cells (HSC) in bone marrow, the same progenitor as for monocytes and neutrophils [91]. After specification in the bone marrow, cDC progenitors are able to proliferate in lymphoid organs and peripheral tissues [92]. The development of all subsets of DC is dependent on the cytokine FMS-like tyrosine kinase 3 ligand (FLT3L) [93] and the differentiation into DC subtypes is controlled by different transcription factors. cDC1 differentiation is controlled by IFN-regulatory factor 8 (IRF8), DNA-binding protein inhibitor ID2, basic leucine zipper transcriptional factor ATF-like 3, and nuclear factor IL-3-regulated protein. cDC2 development depends on RELB, PU.1, recombining the binding protein suppressor of hairless and IRF4 [87]. The transcription factor E2-2 regulates pDC development [94]. Immature DC migrate out of the bone marrow to colonize peripheral tissues, where they encounter antigens, such as danger-associated molecular patterns (DAMPs), which are recognized by pattern recognition receptors (PRR) that are expressed by each subset of DC. Concerning the phenotype of these DC, cDC1 are characterized by CD141, cDC2 by CD1c, and pDC by CD123.

### 3.2. Dendritic Cells in Cancer

In cancer, DC predominantly play an anti-tumorigenic role through the cross-presenting of exogenous antigens via MHC class I and class II to CD8 T cells and CD4 T cells, respectively, and through the secretion of immune-stimulatory cytokines. In a tumor context, cDC1 are both lymph node residents and migratory populations. These cells sample antigens in blood and lymph fluid to deliver them to lymph nodes. In the tumor, these cells are an important source of CXCL9 and CXCL10, which allow for the infiltration of both naïve and pre-activated T cells [95]. They can also produce high levels of IL-12 and type I IFN, which promote DC-mediated cross-priming of anti-tumor T cell responses, but also help to maintain CTL effector functions within the TME [96]. In contrast, cDC2 induce CD4 T cell responses and activate TH17 cells, but they do not deliver antigens to lymph nodes [97]. DC can also derive from circulating monocytes in the context of cancer or inflammation. These moDC, for monocyte-derived DC, can have the same presenting role as the resident cDC1 [98] and produce high levels of IL-15 in order to support anti-tumor T helper cell type I responses [99] or express TRAIL to mediate tumor cell apoptosis [100].

Alongside these anti-tumorigenic capacities, DC can, under certain circumstances, act as pro-tumoral cells in the TME. Despite the capacity of pDC to produce pro-inflammatory type I IFN, their presence can be associated with a poor prognosis in breast cancer, melanoma, and ovarian cancer [101,102]. Indeed, the poor activation of pDC in the TME and their active instruction by tumor cells lead to an immunosuppressive function through the production of IDO, IL-10, or OX40 [103]. The strong interaction of pDC and multiple myeloma (MM) cells induce the secretion of IL-3, which stimulates both pDC survival and MM cell survival and proliferation [104]. moDC in the TME can also be efficient producers of iNOS, TNF-α, IL-6, and IL-10 and hamper T cell proliferation [97,98].

Additionally, DC express the SIRPα receptor and the leukocyte immunoglobulin-like receptor B1 (LILRB1), which, respectively, recognize CD47 and MHC class I-associated β2M subunits at the surface of tumor cells, blocking phagocytosis [105,106]. CD24 expressed by tumor cells can also inhibit phagocytosis by DC after engagement of Siglec-10 [107].

Immune checkpoints, such as CTLA-4 and PD-1, have been shown to negatively regulate the activity of DC. DC can express CTLA-4 and secrete it within microvesicles that could interact with costimulatory molecules, e.g., CD80/CD86 on bystander DC leading to the loss of expression of these molecules and, thus, to the non-activation of CD8 T cells [108]. DC can also express PD-1, limiting the production of the pro-inflammatory cytokines IL-12 and TNF-α during ex-vivo restimulation [109]. The ligation of PD-1^+^ DC to PD-1L expressed by ovarian tumor cells induced the suppression of antigen presentation, costimulatory molecule expression, and pro-inflammatory cytokine secretion [110]. TIM3 also expressed by DC negatively regulates IFN-α production by pDC [111] and CXCL9 production by cDC1 limiting the recruitment of cytotoxic T cells into the tumor [112]. Moreover, the interaction of HMGB1 and TIM3 on DC interferes with nucleic acid recruitment to endosomal compartments impairing the innate immune sensing of nucleic acids released by dying tumor cells [113].

In addition, the tumor can also interfere with the anti-tumorigenic functions of DC. Tumor-derived IL-6 and CSF-1 affect DC differentiation, promoting lineage commitment toward suppressive monocytes [114] and VEGF inhibits DC maturation by suppressing NFκB signaling in hematopoietic progenitors [115] as well as the tumor-derived TGF-β can inhibit antigen uptake [116].

## 4. Neutrophils

Neutrophils play an important role in the innate immune response against pathogens through phagocytosis, the release of anti-microbial peptides/proteases, and release of neutrophil extracellular traps (NETs). However, they are also able to modulate cancer growth and metastatic progression.

### 4.1. Origin and Physiological Roles of Neutrophils

Neutrophils represent 60% of all leukocytes in the bone marrow and their homeostasis is maintained thanks to an equilibrium between granulopoiesis, storage in bone marrow and release, and migration toward peripheral tissues. These cells have a relatively short life, but they are essential for pathogen elimination and represent an important link between innate and adaptive immunity.

In the bone marrow, the common myeloid progenitors are differentiated into common granulocyte monocyte progenitors (GMP), which can either lead to the monocyte lineage, depending on the C/EBP-α, or to the neutrophil and eosinophil lineage, thanks to the C/EBP-ε [117,118]. The acetylation of C/EBP-ε and the lack of expression of GATA-1 then lead to the ultimate differentiation into neutrophils [118]. Granulopoiesis is the formation of granules within neutrophil development; these granules are essential for the neutrophil functions. This process begins at the GMP stage, which progresses through a series of progenitors, including myeloblasts, promyelocytes, and myelocytes, which are able to proliferate, before becoming metamyelocytes that no longer proliferate, leading finally to mature neutrophils [119]. However, neutrophils are also plastic-like macrophages and they appear to be disease- or tissue-specific, and distinct neutrophils subsets can therefore be distinguished through their appearance, their density, or their surface receptor expression profiles. Neutrophils can be polarized in vitro into pro-inflammatory N1 by LPS and IFN- or anti-inflammatory N2 by IL-4 [120]. However, the question of plasticity with the N1/N2 switch remains unclear, knowing that neutrophils have a much shorter life-span than macrophages.

Neutrophils contribute to tissue injury by amplifying the inflammatory response, by the release of toxic effectors and the phagocytosis of pathogens. First, neutrophils have to be recruited in peripheral organs thanks to chemokines (G-CSF, CXCL1, CCL2, CXCL10) that are produced by conventional DC that encounter pathogens [121]. In addition, endothelial cells that are activated through PRR that detect pathogens upregulate P-, L-, and E-selectins that maximize neutrophils recruitment through their capture by endothelial cells, their rolling, and transmigration [122]. Neutrophils express cellular adhesion molecules (ICAM-1, ICAM-2, VCAM-1), which are essential to pass through endothelial junctions [123]. At the site infection, neutrophils are able to internalize microorganisms through Fcγ receptors, C-type lectins, or complement receptors. Following engulfment, primary and secondary granules fuse to the phagosome and release their antimicrobial contents and reactive oxygen species (ROS) to kill the microbe. Neutrophil cytoplasmic granules containing various cytotoxic factors also play an important role in killing pathogens [124]. Under flow conditions in the blood, neutrophils develop an additional mechanism for engaging and capturing circulating pathogens. Upon sensing bacteria in the blood, neutrophils release their DNA in a netlike configuration to create and release traps, called NETs. NETs are covered with elastase, histones, and other toxic molecules that kill pathogens [125]. However, if the release of anti-microbial molecules and NETs are essential for killing pathogens, then this can cause some toxic effects in the surrounding cells leading to tissue injury and thereafter contribute to the development of many non-infectious diseases, such as lung injury or rheumatoid arthritis [126]. Thus, neutrophils functions are tightly regulated. 

Subsequently, neutrophils can be involved in the tissue repair following tissue injury; first, as professional phagocytes to remove tissue debris at the site of injury. Secondly, neutrophils can release growth factors and pro-angiogenic factors that contribute to regeneration and revascularization [127]; and, thirdly, neutrophils become apoptotic and they are cleared by macrophages, which also leads to the release of the tissue-repairing cytokines TGF-β and IL-10 [128].

Therefore, neutrophils have some beneficial effects on the resolution of infection, but they can also have detrimental activities and the unbalance between these effects can favor disease development [129].

### 4.2. Neutrophils in Cancer

In cancer, the dysregulation of granulopoiesis leads to different subsets with specific roles in tumor progression.

Two neutrophil subsets, high-density neutrophils (HDNs) considered as mature, and low-density neutrophils (LDNs), a mixture of mature and immature neutrophils, were identified in various tumor models by differential density centrifugation. The increase of these LDNs in the peripheral blood was associated with tumor growth and metastasis [130]. In addition, the two categories, N1 and N2, were clearly described in cancer through their respective anti- and pro-tumorigenic functions with the concept of plasticity as for TAM, which has not yet been shown in infectious processes. The neutrophils that are found in the TME are referred to as TAN (tumor associated neutrophils). The CXCR2 axis controls the mobilization of neutrophils from blood towards the tumor [131]. When compared to neutrophils in infectious disease, where they have a short life span, TAN display a high longevity and N1/N2 TAN can be converted into each other [132]. TME enriched in TGF-β or G-CSF induces the polarization towards the N2 phenotype, whereas IFN-γ, IFN-β, or LPS lead to N1 polarization [133,134].

The anti-tumor activities of neutrophils have been shown in different contexts and notably in a colon cancer mouse model, in which the depletion of neutrophils was associated with an increase of proliferation, growth, and invasiveness of tumor cells [135]. The mechanisms of anti-tumor functions of neutrophils are varied. Through the generation of ROS, such as H2O2, neutrophils were shown to be able to kill tumor cells in vitro and in tumor mouse models [136]. The production of ROS by neutrophils can also suppress the pro-tumorigenic role of the IL-17-secreting γδ T cells in tumors, which inhibits their proliferation [137]. By inhibiting STAT5, neutrophils are able to induce the apoptosis of prostate cancer cells [138], and their secretion of granzyme B in the TME in colon cancer mouse models leads to the killing of tumor cells [139]. Moreover, neutrophils are able to kill cancer cell in a contact-dependent manner through the interaction of cathepsin G at their surface with the receptor for advanced glycation end products (RAGE) [140]. Thanks to the production of pro-inflammatory factors, such as CCL2, IL-8, CCL3, and IL-6, and their crosstalk with activated T cells, TANs, in early-stage human lung cancer, are actively involved in the stimulation of T cells responses limiting tumor progression [141]. In addition, IFN-γ produced by recruited monocytes in lung tumors activates the encoding of the TMEM173 gene for the protein STING (stimulator of interferon genes) within neutrophils, which stimulates the neutrophil-mediated killing of disseminated cancer cells [142]. Finally, it was shown that infiltration by neutrophils enhances the prognostic significance of colorectal cancer infiltration by CD8 T cells, improving survival in human colorectal cancer [143].

On the contrary, neutrophils display pro-tumorigenic functions that enhance tumor cell proliferation, invasion, angiogenesis, metastasis, and immune suppression. Tumor cell proliferation can be promoted by interaction with neutrophils in hypoxic conditions and mediated by the production of neutrophil elastase [136], as in the case of the acute promyelocytic leukemia progression [144]. The enhancement of non-small cell lung cancer cells proliferation is also favored by elastase that is produced following their interaction with neutrophils as well as the production of immunosuppressive PGE2 [145]. NET overproduction was also involved in promoting tumor cells proliferation, but also migration and invasion, favoring the crosstalk between glioma cells and the TME by regulating the HMGB1/RAGE/IL-8 axis [146] or by activating pancreatic tumor growth through the DNA releases from NET [147].

The role of neutrophils in the metastatic process was shown by the association between neutrophil and circulating tumor cells within the bloodstream in breast cancer patients and mouse models [148]. By a proteomic approach, the iron-transporting protein transferrin was identified as the major mitogen for tumor cells that are secreted by neutrophils and the depletion of neutrophils suppressed transferrin production and lung metastasis [149]. Neutrophils have also been suggested to promote cancer cell adherence and, thereby, mediate metastasis in a murine model of liver metastasis [150]. Finally, human neutrophils were shown to induce tumor cell migration and interact with melanoma cells via β2 integrin [151].

Neutrophils also play an important immunosuppressive role in impairing T cells proliferation and the activation in the TME in ovarian cancer for instance [152]. A high mature neutrophils/T cell ratio in multiple myeloma patients was correlated with a weak progression-free survival that was associated with an immunosuppressive profile of the infiltrated neutrophils [153]. In advanced stages of primary melanomas, TAN were shown as expressing PD-L1, CXCR4, CCR5, Adam17, and Nos2, leading to the immunosuppression of T-cell proliferation [154]. The expression of PD-L1 by TAN was correlated with the induction of PD-1 on CD8 T cells and their in vivo depletion delayed tumor growth with a significant increase of the frequency of proliferating IFN-γ-producing CD8 T cells [155]. Besides, the expression of PD-L1 by neutrophils can be promoted by IL-6 that is produced by human ovarian tumor cells under the effect of long non-coding RNA [156]. Finally, neutrophils are also able to suppress NK cell cytotoxicity, which results in defective antitumor responses and promotes metastasis in mice [157].

This balance between pro- and anti-tumor effects shows the plasticity of neutrophils, depending on different factors and cell contacts in the TME.

## 5. MDSC

MDSC form a heterogeneous group that consists of activated immature myeloid cells at different stages of hematopoiesis, which are able to exert strong anti-inflammatory response. In contrast to previously discussed cell types, MDSC presence in healthy individual is limited, as they are a product of chronic inflammation. In the context of cancer however, they are stated as one of the main actors in immune-escape and maintenance of immunosuppressive TME and their presence is repeatedly connected with failure in response to ICI therapies [158].

### 5.1. Origins and Roles of MDSCs

By nature, these cells are the result of a disrupted hematopoiesis and they are usually discussed in the context of cancer. However, they are gaining increasing recognition in physiological inflammation and aging, injury, trauma, and systemic infections, such as sepsis. MDSC generation is aimed at restraining hyper-inflammation and protecting the host from autoimmunity [159]. A good example of this phenomenon can be observed in severe COVID-19 cases, where there is a substantial increase in the percentage of MDSC in peripheral blood of patient up to 90% of all PBMC (normally MDSC constitute around 1% of all PBMC in healthy donor), which again decreases upon the end of infection [160]. MDSC consist of two main groups of cells: polymorphonuclear (PMN-MDSC) initially called granulocytic (G-MDSC) and monocytic (M-MDSC) [161]. Further research enabled the distinction of the third group, which is represented by even less specialized cells, termed as early-stage MDSC (eMDSC) [162]. Normally, immature myeloid cells reside in the BM and they leave this site when they reach a certain maturation stage. Nonetheless, in the presence of stress signals, they are recruited from the BM in their immature state [163]. High levels of cytokines, such as: GM-CSF, G-CSF, and M-CSF, in conjunction with IL-6, IL-1β, VEGF, and FGF-β produced by stromal or tumoral cells, are responsible for a disturbance of hematopoiesis and expansion of MDSCs. GM-CSF and M-CSF are crucial in the case of M-MDSC, whilst G-CSF is a key cytokine for PMN-MDSC. The phenotypical characterization of MDSC and their distinction from mature myeloid cells can be a serious problem and often requires the comparison of multiple surface markers and the performance of functional assays [164]. In mice, MDSC can be characterized as Gr-1^+^CD11b^+^ cells and, further, M-MDSC were described as CD11b^+^Ly6C^high^Ly6G^−^ cells, while PMN-MDSC as CD11b^+^Ly6C^low^Ly6G^+^ expressing cells [165,166]. In humans, a great number of surface phenotypes have been described with high variations between individuals. The mouse equivalent of PMN-MDSC was defined as CD11b^+^CD14^−^CD15^+^ or CD11b^+^CD14^−^CD66b^+^CD33^DIM^, whereas M-MDSC are CD11b^+^CD14^+^CD15^−^HLADR^−/low^CD33^+/high^ [165,167,168]. eMDSC were defined as Lin^−^(CD3, CD14, CD15, CD19, CD56) HLADR^−^CD33^+^ [164]. Additionally, the use of more recent markers allowed to further distinguish MDSC from their mature counterparts. For example M-MDSC are S100a9^high^, whilst mature monocytes are S100a9^low^ [169,170]. In the human PMN population, the LOX1 marker turned out to be expressed on PMN-MDSC in contrast to mature neutrophils [171,172]. Additional to the phenotypic features detailed above, mouse studies showed that CD84 expressed on PMN-MDSC helps to distinguish them from neutrophils [173]. It is worth mentioning that the PMN-MDSC population is partially enriched in a mononuclear cell fraction during centrifugation in the Ficoll gradient, due to the decreased granularity of those cells [164]. In humans, MDSC can therefore be distinguished from neutrophils and monocytes based on their phenotypic markers, density gradient, but also by their unique functionality. Physiologically myeloid activation subsides quickly after the cessation of stimuli; that said, MDSCs, as a product of chronic inflammation, show continuous activation with an impaired ability to produce pro-inflammatory factors. MDSC are also characterized by very limited phagocytic properties. Instead, they excel in the field of immune suppression [171]. The repression of the immune response by MDSC takes place through a wide range of different mechanisms. In humans, PMN-MDSC are mostly known to inhibit T cells via the production of ROS [174]. M-MDSC mediate T cell suppression through the inducible nitric oxide synthase (iNOS) generation of NO, release of suppressive cytokines, such as IL-10, and prostaglandin PGE2 [175,176,177]. For instance, iNOS play an essential role in MDSC-mediated T cell suppression inhibiting antigen specific T cell proliferation in mice with proteoglycan-induced autoimmune arthritis [178]. PMN-MDSC also expressed immune checkpoint receptors, such as PD1, which can be upregulated in multiple sclerosis and leads to T cell inhibition [179].

### 5.2. MDSCs in Cancer

Initial remarks on the presence of MDSC were made at the beginning of the 20th century upon the characterization of tumorigenesis. With time, it has been shown that the numerous myeloid cell populations detected in different cancers were able to negatively impact T cell proliferation and activation. Moreover, it has been observed that these cells lack some of the markers of immune cell populations, such as T cell, NK cell, or macrophages, and, therefore, were initially named Null cells or Natural suppressor cells (NS) [163]. Only recently the term Myeloid Derived-Suppressor Cells was introduced to more appropriately represent the nature and functionality of these cells [163,180]. In humans, single cells RNA experiments of pancreatic lesions revealed an increased infiltration of MDSC, depending on the severity of the disease [181]. Indeed, MDSC can constitute up to 51% of non-epithelial cells in Pancreatic Ductal Adenocarcinoma (PDAC) when compared to 5% in non-invasive Intraductal Papillary Mucinous Neoplasms (IPMNs) correlated with a loss of cytotoxic T cells in early lesions and a gain of immunosuppressive myeloid populations in progressing diseases. A multicenter analysis showed that PMN-MDSC are a leading population of MDSC in solid tumors and their expansion is especially intensified in the context of tumorigenesis when compared to other inflammatory states [172]. Interestingly, mouse research showed that, in TAM rich tumors, M-CSF/M-CSFR signaling indirectly blocks the recruitment of PMN-MDSC to the tumor site. This phenomenon was observed after anti CSF-1R therapy, which causes increased infiltration of the tumor by PMN-MDSC (and it is stated as one of the reasons for failure of CSF-1R targeted monotherapy) [182]. 

The MDSC differentiation program is associated with abnormal STAT3 signaling and the inhibition of the IRF 8 branch, leading to sustained cell proliferation and the inhibition of their terminal differentiation. STAT3 activation also leads to the expression of S100A9, which plays an essential role in MDSC biology. It is important in the stabilization of the STAT3/C/EBPb complex, which is crucial for the blocking of terminal differentiation and maintenance of the immune-suppressive function of MDSC [183,184]. For example, S100A9 KO mice exhibit a very potent antitumor response that is reversed upon the transfer of S100A9 positive MDSCs [185]. Furthermore, it has been shown that the administration of S100A9 protein alone to KO mice can restore the MDSC phenotype [184]. 

MDSCs are able to suppress both the innate and adaptive immune systems through various mechanisms. MDSC suppress T cell functions by producing ROS and RNS inducing the nitration of TCR and MHC-I, which hinders peptide recognition and disrupts T cell IL-2 signaling [174,186]. MDSC can be involved in the depletion of amino acids that are essential for T cell functions and proliferation. MDSC expression of the arginase 1 leads to the depletion of L-arginine, an important nutrient for T cell and NK cell proliferation [187]. By sequestering cysteine, MDSC limit the availability of this amino acid in the microenvironment, thereby hindering T cells activation and functions [188]. Tryptophan, which is also essential for T cell survival and functions, can also be depleted by MDSC through the expression of IDO that is involved in the catabolism and degradation of tryptophan [189]. Furthermore, CD8 T cells can be inhibited or paralyzed, as stated by the authors, by MDSC in the TME through expression of dicarbonyl radical methylglyoxal [190]. The intratumoral migration of T cells can also be impaired by MDSC through the peroxynitration, which alters CCL2 levels or through their plasma membrane expression of ADAM17 (a disintegrin and metalloproteinase 17) that cleaves the ectodomain of L-selectin (CD62L) on T cells [191,192]. MDSC have also been shown to impair NK cell functions. They can block the IFN-γ production by NK cells by affecting Stat5 activity and reducing NKG2D expression by NK cells through cell–cell interactions, leading to the suppression of NK cell cytotoxicity in tumor-bearing mice [193,194]. The production of PGE2 by MDSC was shown to enhance the stemness of uterine cervical cancer and the promotion of PD-L1 expression in epithelial ovarian cancer [195,196].

Moreover, MDSC can drive immune tolerance to tumors by increasing the number of Treg in the tumor. This is related to the interaction of the immune stimulatory receptor CD40 on MDSC with the CD40L that is expressed on Treg, and indirectly to the secretion of IL-10, TGF-β by IFN-γ stimulated MDSC [197,198]. 

Finally, MDSC confers resistance to immune checkpoint inhibitors, as MDSC depletion was shown to enhance the efficacy of anti-PD1 and anti-CTLA4 treatment with a complete tumor regression and a decrease of metastasis in an aggressive breast tumor mouse model [199]. Moreover, through the expression of Arg-1, MDSC confers resistance to bortezomib in human myeloma cell lines [200], which can be reversed while using Arg-1 inhibitors that were shown to decrease the L-arginase activity in correlation with a reduction of tumor growth in multiple mouse models of cancer [187].

Therefore, MDSC also play an important role in the resistance of tumor cells against the immune system to certain therapies.

## 6. Mast Cells

Mast cells (MCs) are innate immune cells that are located in virtually all tissues and they are particularly numerous in barrier tissues, such as skin and mucosa. They are characterized by the co-expression of CD117 (KIT) and FcεRI. They constitute a versatile population of sentinel cells that are endowed with multiple immune defenses and regulation capabilities, such as: defense against parasites and micro-organisms, defense against venoms, initiation of the inflammatory vascular phase, interactions with CD4 and CD8 T cells, and positive or negative modulation of the immune response.

### 6.1. Origin and Physiological Roles of Mast Cells

MC precursors are produced in the bone marrow through the stem cell factor (SCF)-driven differentiation of hematopoietic stem cells [201,202]. These precursors circulate in the blood and are home to tissues thanks to the expression of α4β7 integrins and positive chemotaxis for SCF [203]. MCs differentiate globally into two subsets, depending on the type of tissue they colonize and its microenvironment. Homing to the mucosa drives the differentiation of mucosal mast cells (MMCs in mice) or MCT (in humans, for tryptase expressing MC). Homing to connective tissue or serosa (such as skin or peritonea) drives the differentiation of CTMCs (connective tissue MCs in mice) or MCTC (in human, for tryptase and chymase expressing MCs). CTMC maturation is driven by SCF, NGF, and IL-9 provided by neighboring cells, such as epithelial cells or immune cells. 

It was recently reported in a murine model that, like macrophages, mast cells have dual developmental origins, which arise from primitive and adult hematopoiesis [204]. Largely of yolk sac origin in early life, they are progressively replaced by mast cells that are derived from adult HSCs. 

Like granulocytes, MCs store several bioactive molecules in their granules (such as histamine, tryptase, chymase, chemokine, TNF) that can be swiftly released upon activation by degranulation [205]. Bioactive components within the granules are embedded in a matrix of proteoglycan that allows for their storage and regulation of their biological activity [206]. Upon degranulation, the matrix is exteriorized and it acts as a bio-diffusor that take part in regulating half-life and bio-activity of the mediators [207]. MC also produce neosynthesized mediators such as eicosanoids (PGD2, PGE2, Cysteinyl leukotrienes), chemokines (CXCL8, CCL2, CCL3, CCL4) and cytokines (IL-5, IL-6, TNF, IL-13, IL-10, IL-1β). Finally, MCs can play the role of antigen presenting cells [208,209,210]. 

MCs are abundant in barrier tissues that are exposed to the external environment. They play the role of surveillance outposts that sense their environment. Indeed, MCs respond to both innate and adaptive immunity stimuli and to changes in tissue homeostasis [211,212]. MC express several PRRs, which allows them to sense pathogens or danger signals [213]. Moreover, MC biological responses are deeply impacted by the alarmins IL-33 and TSLP [214]. They strongly express the IL-33 receptor ST2, allowing for them to sense stressed or dying cells in their neighborhood. IL-33 swiftly potentializes several MC functions, such as degranulation or cytokine production [215,216]. MCs indirectly recognize antigens via Abs/Fc receptors. MCs express Fc receptors for IgE and IgG, depending on the subset. The expression of the high affinity receptor for IgE is a hallmark of mast cells, enabling interactions with IgE-targeted antigens and their involvement in immediate hypersensitivity reactions. MCs also express activatory and inhibitory (in rodents) receptors for IgG [217]. For instance, IgG-opsonized pathogens are recognized by human MCs via FcγRIIA and they induce a polarized degranulation, called ADDS (antibody-dependent degranulatory synapse). Finally, connective tissue MCs express a receptor for cationic secretagogues, MRGPRX2 in humans (the ortholog of Mrgprb2 in rodent) [218]. This receptor recognizes a vast array of cationic compounds, such as neuropeptides (Substance P, Vaso-intestinal peptide), antimicrobial peptides (cathelicidin, human α-defensin2), or bacterial quorum sensing molecules [219], and triggers MC degranulation and cytokine production [207,220,221]. 

The production of such a large array of receptors and mediators endows MCs with the unique ability to detect tissue damages and interact with vascular, immune, and nervous systems. MCs can exhibit direct microbicidal activity [222]; nevertheless, it seems that the principal role of MCs against pathogens is to trigger an alarm and organize the inflammatory response. This task is enabled by their strategic location in tissues: near blood vessels, nerve endings, and lymphocyte-rich areas [222]. 

MCs are responsible for the main vascular changes that accompany the inflammatory response, thanks to the production of potent vasoactive compounds, such as histamine, TNF, and tryptase. They take part in the recruitment of immune effector cells in inflamed tissues by producing several chemokines. Several studies have reported the close proximity between MC and nerve fiber in skin and intestine [223,224,225]. It appears increasingly clearly that MCs and sensory neurons form a functional unit in the skin and the gut, and that nerve cell-MC crosstalk is an important functional module in the response to tissue aggressions [210,226,227]. Moreover, MCs are found near T cell rich areas in tissues and they can interact with several T cell subsets, such as CD4 Th cells [209], CD8 T cells [228], and γδ T cells. Upon IFN-γ stimulation, MCs express MHC class II and costimulatory molecules and were shown to influence Th cell cytokine production in vitro [229,230]. Likewise, cognate and non-cognate crosstalk between MC and CD4 T cells were shown to mold both MC and T cell responses [230,231,232,233].

Finally, MCs also contribute to tissue repair, matrix remodeling, fibrosis [234], and wound healing [235]. MC proteases, such as tryptase and chymase, can inactivate inflammatory factors and avoid excessive tissue damage [236], and participate directly or indirectly in tissue remodeling [237].

In conclusion, MCs are versatile immune cells that take part in numerous processes of the immune response and are involved in several pathophysiological mechanisms [238,239]. A partial understanding of their complex role is especially apparent in the context of cancer.

### 6.2. Mast Cells in Cancer

Because of their natural presence in all tissues, MCs are immune components of the TME and are de facto present there from the first stage of carcinogenesis. Nevertheless, their role in the cancer pathophysiology remains elusive. 

Numerous studies (reviewed extensively in [240]) have reported that MCs can adopt different roles toward cancer cells, depending on the type of tumor, stage of the disease, and their localization in TME. All of those conditions dictate whether MCs will display pro- or anti-tumoral properties or will simply remain bystanders. A good example of contribution of MCs to tumor development dependent on the stage of the disease was described in prostate cancer, where MCs play a pro-tumorigenic function at an early stage, but became dispensable at a later stage [241] or in NSCLC where peritumoral MC density is of favorable prognostic in stage I, but not in stage II [242]. Recently, it was highlighted that not only the density, but also micro-localization, of MCs can modify their involvement in cancer development. Differences in the pro or anti-tumorigenic role of MCs according to their localization have been reported at least in Melanoma [243], lung [244], and prostate cancer [245,246]. In lung cancer, the infiltration of MCs inside tumor islets was associated with a better outcome independent of tumor stage [244]. In prostate cancer, intratumoral MC density was associated with a good prognosis [246], whilst high peritumoral MC density was described to promote tumor growth [245]. 

One of the clearest functions of MCs in TME is their participation in angiogenesis through VEGF secretion. MC density was positively correlated with angiogenesis in Melanoma [247,248], colorectal, lung [249], and pancreas cancer [250]. Interestingly, a recent study showed that VEGF production by MCs can be directly triggered by cancer cells-derived extracellular vesicles [251].

MCs have also been described to take part in epithelial to mesenchymal transition (EMT). In thyroid cancer, Visciano and colleagues showed that MCs promote EMT via CXCL8 production, leading to AKT phosphorylation and Slug expression by thyroid cancer cells [252]. The involvement of MCs in EMT was also suggested in a pre-clinical model of lung metastasis. Mice lacking MCs (Mcpt5-Cre) showed a reduced melanoma colonization in the lung. This observation is associated with a higher expression of E-cadherin and reduced expression of Twist in MC deficient mice when compared to control mice, indicating a change in epithelial/mesenchymal orientation [253]. MCs also mediate malignant pleural effusion via tryptase and IL-1β release after recruitment via tumor derived CCL2 and activation by osteopontin [254]. 

Other studies have suggested a possible role of MCs inside the TME through their ability to interact with other immune cells and to favor or suppress immune responses. In a subcutaneous cancer model, Huang and colleagues showed that SCF-activated MCs can intensify immunosuppression by increasing Treg frequency and secreting adenosine that acts directly on both T cells, by reducing their proliferation, and on NK cells, by decreasing IFN- production [255]. 

Interestingly, MC/T cells cooperation inside the TME was described as an important factor in predicting responses to neoadjuvant chemotherapy in inflammatory breast cancer [256]. The characterization of the tumor microenvironment revealed that having a lower pretreatment MC density was significantly associated with achieving a complete pathological response to neoadjuvant chemotherapy. Moreover, spatial analysis revealed a close proximity between CD8 T cells and MCs when a complete pathological response was not achieved, highlighting MC as an interesting therapeutic target that could improve current therapeutic strategies. Targeting MCs to improve therapeutic strategies was also suggested in a pre-clinical model of melanoma, where Kaesler and colleagues identified MCs accumulating in and around melanomas after anti-CTLA-4 treatment and showed that effective melanoma immune control was dependent on LPS-activated MCs that secreted CXCL10, which promoted the recruitment of effector T cells. They highlighted a new way to target MCs and to involve them in the tumor immune defense [257]. 

These observations suggested different mechanisms by which MCs can impact the TME and influence tumor development. Nevertheless, many other mediators commonly produced by MCs play a role in tumor development, such as: PGE2 [258], IL-13 [259], histamine [260], and TNF [261]. Whether MCs produce these mediators inside the TME remains to be elucidated.

## 7. Eosinophils and Basophils

Just as mast cells, eosinophils and basophils are specialized effector cells playing key roles in the defense against parasites and hypersensitivity type I reactions. These cells share typical receptors and cytokines, but display specific effector functions. Basophils can be found in the bloodstream of healthy individuals and they are rapidly recruited within tissues in the presence of inflammation. Eosinophils circulate and they are resident in the hematopoietic and lymphoid organs, being ready to migrate to the site of allergic reactions. In addition, these cells can also play some pro-tumor roles.

### 7.1. Origin and Functions

Basophils stem from the differentiation of GMP in the bone marrow and then circulate [262]. Eosinophils also differentiate in the bone marrow from IL-5 receptor alpha progenitors and then migrate into the bloodstream [263], and the presence of intracellular acidophilic granules discriminates them from basophils and neutrophils. IL-3 is the most important growth and activating cytokine for human and murine basophils that are produced by the inflamed tissue, but also in an autocrine manner [264]. Basophils, such as mast cells, express the Fcε Receptor I, which has a high affinity for the immunoglobulin E (IgE). IgE is produced during type I hypersensitivity reactions that are observed in various allergic diseases (e.g., asthma, sinusitis, rhinitis, food allergies, chronic urticarial, and atopic dermatitis), but also in the defense against parasites (protozoa and helminths) [265]. Upon activation through the FcεRI, basophils release their granules content, being composed of inflammatory mediators into the environment. These mediators comprise histamine, protease, cytokines, and chemokines, which will activate other inflammatory cells, but also vessels and smooth muscle [266]. Amongst these cytokines, IL4 and IL-13 are potent mediators of the type 2 immune response. Additionally, basophils produce IL-6 and TNF-α [267].

On the contrary, eosinophils weakly express FcεRI, but express other cell surface molecules that are involved in their activation, such as FcγRIIA (receptor for IgG), FcαRI (receptor for IgA), complement receptors, cytokine receptors, especially receptors for IL-3, IL-5, and GM-CSF, chemokine receptors (CCR1 and CCR3), but also some adhesion molecules and TLR7/8 [268]. Through their expression of the ST2 receptor, eosinophil differentiation can be stimulated by IL-33 during inflammatory responses in an IL-5-dependent manner [269]. Several receptors that are present at the surface of eosinophils, upon stimulation, mediate piecemeal degranulation. However, the molecules released depends on the stimulus, as eosinophils are able to secrete cytokines mediating opposite effects. For instance, IFN-γ stimulation induces the secretion of IL-1, whereas stimulation with eotaxin (CCL11) leads to the secretion of IL-4, despite the fact that these two interleukins are stored in the same granules [270,271]. As for neutrophils, when exposed to bacteria, eosinophils are able to release mitochondrial DNA in order to form an extracellular trap containing granule proteins, eosinophil cationic protein, and eosinophil major basic protein 1 that bind and kill bacteria, for instance in inflammatory skin diseases [272]. Moreover, eosinophils also express several inhibitory receptors, such as FcγRIIB, ILT5/LIR3, CD33, Siglec-8/10, p140, and IRp60/CD300a (reviewed in [273]). Eosinophils are thus complex cells that can be either stimulated or inhibited in their proliferation, survival, or functions through these inhibitory receptors.

### 7.2. Basophils and Eosinophils in Cancer

The presence of basophils and eosinophils was detected in several types of tumors. Taking under consideration plasticity and a range of factors, these cells can express it is impossible to unambiguously define their role in cancer. Depending on the circumstances, they can show anti-tumoral effects or favor angiogenesis, cancer cell invasiveness, and maintenance of an immunosuppressive environment. 

In a mouse model of breast cancer, a low level of circulating basophils has been correlated with an increased number of pulmonary metastases [274]. In addition, a study of over 400 women that were diagnosed with colorectal cancer revealed that eosinophil infiltration of the tumor, particularly in the stromal tissue, was associated with a decreased mortality rate [275]. Eosinophils were also found to be potentiators of anti-CTLA4 therapy in breast cancer patients, being correlated with their level of accumulation within the tumor [276]. A direct anti-tumor effect of eosinophils in a melanoma mouse model was shown to be dependent on the presence of IL-33 in the TME [277]. High relative circulating basophils and activated infiltrated basophils were positively associated with improved outcome in melanoma or ovarian cancer patients and, according to several data, basopenia was associated with a poor prognosis in colorectal cancer [278,279,280].

However, these two types of cells could also favor tumor development. Basophils are preferentially circulating, but some lung-resident basophils exhibit a specific phenotype that is involved in the development of alveolar macrophages and their polarization towards a pro-tumor M2 state. Therefore, basophils may be involved in regulating the activity of TAM in the TME [281]. Moreover, recruitment of basophils in tumor-draining lymph nodes of pancreatic ductal adenocarcinoma patients has been shown to be activated by T-cell-derived IL-3 to produce IL-4, inducing a tumor-promoting Th2 inflammation [282]. Eosinophils are also able to exert an IL-4 mediated immunomodulatory function, which induces the switch from a Th1 immune response to a Th2 one. In the same way, it has been postulated that basophils recruited within the skin, in an inflammation-driven model of epithelial carcinogenesis, could promote tumor development via their FcεRI signaling [283].

Alongside their role in the immunomodulation, basophils and eosinophils are able to act on angiogenesis. Infiltrating basophils in human nasal polyps contain VEGF-A localized in secretory granules, which can be released by their IgE-mediated activation. Moreover, these cells also express on their membrane VEGFR-2 and the co-receptors NRP1 and NRP2 involved in the infiltration of basophils at the site of chronic inflammation, such as tumor site [284]. Other angiogenic factors, such as angiopoietins or HGF, can be released by basophils upon activation [285,286]. Some lipids mediators, such as cysteinyl leukotrienes (CysLT), produced by activated basophils [287], also display proangiogenic activities and can activate the expression of the CysLT2 receptor in tumor blood endothelial cells [288]. Eosinophils present in hypoxic tumor microenvironments could also play pro-tumorigenic roles by promoting tumoral angiogenesis via the release of VEGF-A, IL-8, and osteopontin [289,290,291]. 

Finally, by sequestering circulating tumor cells, NET was clearly shown to promote cancer metastasis in murine models and in humans [292]. The extracellular trap that is produced by activated basophils and eosinophils could also have some impact on tumor growth and metastasis, but this remains to be investigated.

Basophils and eosinophils display a plastic phenotype with numerous factors that can be involved in pro-tumor processes and certainly in the resistance of tumor cells to immune cells or to chemotherapies as yet to be investigated.

## 8. Conclusions

In cancer, myeloid cells form a diverse group and are able to adapt their phenotype to the TME, depending on cell interaction, oncogenic drivers, altered metabolism, hypoxia, and various secreted factors. Therefore, these plastic cells are able to both initiate or suppress anti-tumor immune response. Whether macrophages or DC or neutrophils or MDSC or mast cells or eosinophils and basophils, all of these cells have an important role in shaping tumor cells to be resistant against apoptosis, immune cells attacks, or therapeutic agents, and, therefore, to grow and migrate for metastasis formation. Therefore, targeting these cells in the TME is a good opportunity to find new specific targeting therapies or to enhance current therapies by therapies combination.

## Figures and Tables

**Figure 1 cancers-13-00165-f001:**
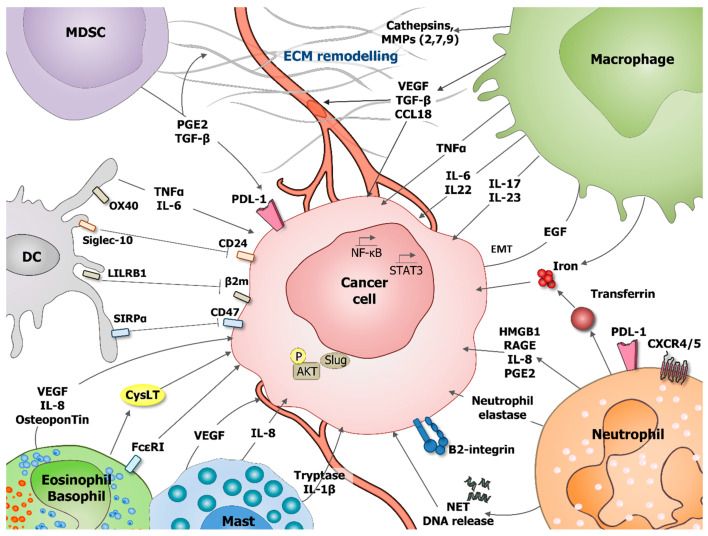
Role of tumor-associated myeloid cells in cancer cells survival, proliferation and migration. During tumorigenesis various myeloid cells populations, including: dendritic cells (DC), myeloid-derived suppressor cells (MDSC), macrophages, neutrophils, eosinophils, basophils, and mast cells can support cancer cells survival, proliferation, and migration. These processes can be stimulated by direct effect on tumoral cells or indirectly by influencing tumor microenvironment (TME), including extracellular matrix (ECM) remodeling and angiogenesis stimulation. Direct effects are mediated through production of interleukin IL-6, IL-8, IL-17, IL-22, IL-23, prostaglandin E2 (PGE2), transforming growth factor beta (TGF-β), vascular endothelial growth factor A (VEGF-A), osteopontin, and tumor necrosis factor α (TNF-α). Neutrophils secrete the iron-transporting protein transferrin which is a major mitogen for tumor cells and release of neutrophil extracellular traps (NET), including their deoxyribonucleic acid (DNA). Neutrophils produce neutrophil elastase favoring tumor cell proliferation and regulate the HMGB1/RAGE/IL-8 axis favoring the crosstalk between glioma cells and the TME. Mast cells release tryptase and IL-1 beta (IL1-β) mediating malignant pleural effusion. Basophils express Fcε Receptor I, promoting their tissue infiltration and producing cysteinyl leukotrienes (CysLT), allowing for proangiogenic activity of activated basophils. DC express OX40, Siglec-10, leukocyte immunoglobulin-like receptor B1 (LILRB1), and SIRPα, which, respectively, recognize OX40 ligand (OX40L), CD24, MHC class I-associated β2M subunits, and CD47 at the surface of tumor cells blocking phagocytosis. Macrophages are an important source of various metalloproteinases (MMPs, MMP2, 7, 9) and cathepsins that provide conduits for tumor cells in the extracellular matrix (ECM). VEGF that is produced by myeloid cells is a major stimulator of angiogenesis.

**Figure 2 cancers-13-00165-f002:**
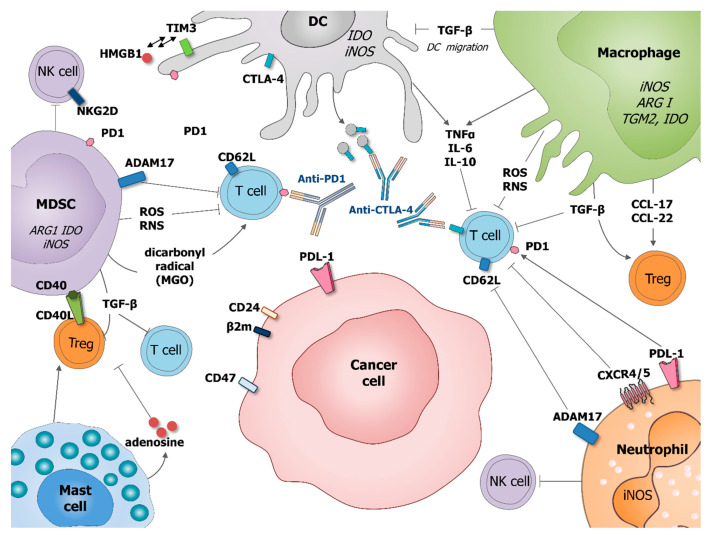
Role of tumor-associated myeloid cells in cancer cells immune-escape and therapy resistance. Macrophages and MDSC produce transforming growth factor beta (TGF-β) which inhibits DC migration at the tumor site, promote regulatory T cells (Treg) and block T cell activation. Macrophages potentiate Treg activation by production of chemokines CCL-17 and CCL-22. DC express immune checkpoint receptors, such as cytotoxic T-lymphocyte-associated protein 4 (CTLA-4), which can be released on the surface of microvesicles that could block costimulatory molecules, such as CD80/86. DC express also programmed cell death protein 1 (PD-1) and T cell immunoglobulin and mucin domain-containing protein 3 (Tim-3) interacting with HMGB1. MDSC suppress T cell functions by producing ROS and RNS inducing the nitration of TCR and MHC-I, as well as producing dicarbonyl radical methylglyoxal in the TME inhibiting CD8 T cells. MSDC express CD40 interacting with is ligand CD40L present on the surface of Treg. Mast cells can stimulate Treg numbers and secrete adenosine, which inhibits T cell proliferation. Neutrophil, as MDSC, expresses a disintegrin and metalloproteinase 17 (ADAM17) that cleaves the ectodomain of L-selectin (CD62L) on T cells. Neutrophil and cancer cells might express PD1 ligand (PDL-1) which inhibits activation of T cells expressing PD1. Neutrophils express CRCR4/5 leading to the immunosuppression of T-cell proliferation. MDSC and neutrophils, are also able to suppress NK cell cytotoxicity. By diminishing the response of various immune cells, tumor-associated myeloid cells can also negatively influence outcome of anti-cancer therapies, especially various immunotherapies.

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
