# Peer review of "Cancer Cells Resistance Shaping by Tumor Infiltrating Myeloid Cells"

_cancers, 2021, doi:10.3390/cancers13020165_

Round 1

Reviewer 1 Report

The manuscript by Domgala et al, is an very nice effort highlighting the role of tumor infiltrating myeloid cells in cancer resistance. 

The review article very nicely summarizes the role of different tumor associated myeloid cell. However, one of the major cancer is that the review article is very descriptive and hard to follow. For example lines 97-117 or so there is one sentence about MPs in context of different organs without getting into any depth. Similarly in other sections there is a need to simplify the content. Authors should consider a table summarizing the role, impact and therapeutic potential of myeloid cells.

Figures 1 and 2 are very nice depiction of the underlying mechanisms.

Review can benefit immensely from some scientific editing.

Author Response

I would like to thanks the reviewer 1 for his comments and suggestions.

We tried to simplify the content in all the sections of the text and did a thorough proofreading.

I hope that our work will satisfy the reviewer.

However, we think that a table is not necessary in addition to the two figures which summarize the protumoral effect of the different myeloid populations.

Reviewer 2 Report

The review article presented by Marcin Domagala et al. focus on the immune cells and particularly myeloid cells which can infiltrate the tumor and shape cancer cells to resist apoptosis, immune attacks and treatments. The Authors clearly summarize and discuss the characteristic (origin, physiological and pathological roles) of the various tumor infiltrating myeloid cells including macrophages, dendritic cells, neutrophils, myeloid-derived suppressor cells, mast cells or eosinophils and basophils. These cells act alone or in concert to shape tumor cells resistance through molecular interaction or secreted factors favoring survival, proliferation and migration of tumor cells but also immune-escape and therapy resistance.

The Authors conclude that reorganizing the myeloid landscape can clear away the roadblock to a successful cancer therapy.

The review is well written and include a balanced, comprehensive and critical view of the research area.

Minor points:

  1. -The Authors must carefully review references section. A lot of information is missing:
  • Ref. 7, 28, 57, 68, 165, 189, 190, 206, 303: the number of page is missing
  • Ref. 62,65,67: the last page number is missing
  • Ref. 316: the number of volume and page is missing
  • Ref. 285, 287: doi number is missing
  • The authors should uniform how write the name of journals; sometimes they are written in extenso and sometimes are abbreviated.
  1. Line 757 and 759: insert the paper of (Johansson MW Front Med 2017) and the paper of (Johnston LK JI 2016) in the references, and in the text insert the relative reference number.
  2. Line 32, 192, 196, 221, 312, 336, 386, 603, 610, 645, 661,753, 755, 762: somethings is missing
  3. Line 239- CRYAB: explain the meaning
  4. Line 451- TMEM173/STING: explain the meaning
  5. Line 447- MCP-1, MIP-1α: use the current nomenclature for these chemokines
  6. Correct some typos errors

Author Response

I would to thanks a lot the reviewer for his comments and suggestions.

We did all the corrections on the text and references (pages missing, abbreviations...)

We corrected also all the typos errors.

Reviewer 3 Report

The authors present a complete and detailed overview of the role of cell of the myeloid compartment in shaping cancer cell resistance. The review draws a complex picture of cell interactions. The text is very dense and somewhat long.

Although the review focuses on the description of the myeloid cell network, I would have wished to have some information on how to tackle these cells in the treatment of cancer. Some examples on what has already been done or options of therapeutic strategies might give the text even more attractive.

In addition, the two figures are like the text: complete and dense. They could be divided into several figures depending on the type of cells involved. This is only a suggestion.

Details

Line 194, the verb “phagocytose” should be replaced by “phagocytize”

Author Response

I would like to thanks the reviewer for his comments and suggestions.

As the reviewer suggested, we simplified and  shortened the text.

We did not deliberately detail the treatments in cancer targeting the cells discussed in this review because that was not the goal. Other very recent reviews in the litterature as one of our team (Laplagne C et al, Int J Mol sci: Latest advances in targeting the tumor microenvironment for tumor suppression), addressed already that.

I understand the point made about the density of the 2 figures, however, we believe that we may lose some information if we divide them. Additionally, the other two reviewers said the figures were fine.

I hope that the new work made on the text will satisfy the reviewer.

Round 2

Reviewer 1 Report

Authors have addressed concerns from the previous review